# Postnatal Overfeeding during Lactation Induces Endothelial Dysfunction and Cardiac Insulin Resistance in Adult Rats

**DOI:** 10.3390/ijms241914443

**Published:** 2023-09-22

**Authors:** Antonio Tejera-Muñoz, Lucía Guerra-Menéndez, Sara Amor, Daniel González-Hedström, Ángel Luis García-Villalón, Miriam Granado

**Affiliations:** 1Research Support Unit, Hospital General La Mancha Centro, 13600 Alcázar de San Juan, Spain; antoniotemu@gmail.com; 2Instituto de Investigación de Castilla-La Mancha (IDISCAM), 45071 Toledo, Spain; 3Departamento de Ciencias Médicas Básicas, Instituto de Medicina Molecular Aplicada (IMMA) Nemesio Díez, Facultad de Medicina, Universidad San Pablo-CEU, CEU Universities, Urbanización Montepríncipe, 28660 Boadilla del Monte, Spain; lguerra@ceu.es; 4Departamento de Fisiología, Facultad de Medicina, Universidad Autónoma de Madrid, 28029 Madrid, Spain; sara.amor@uam.es (S.A.); dgonzalezhedstrom@gmail.com (D.G.-H.); angeluis.villalon@uam.es (Á.L.G.-V.); 5CIBER Fisiopatología de la Obesidad y Nutrición, Instituto de Salud Carlos III, 28029 Madrid, Spain

**Keywords:** cardiovascular, aorta, insulin, obesity, overweight, childhood obesity

## Abstract

Early overnutrition is associated with cardiometabolic alterations in adulthood, likely attributed to reduced insulin sensitivity due to its crucial role in the cardiovascular system. This study aimed to assess the long-term effects of early overnutrition on the development of cardiovascular insulin resistance. An experimental childhood obesity model was established using male Sprague Dawley rats. Rats were organized into litters of 12 pups/mother (L12-Controls) or 3 pups/mother (L3-Overfed) at birth. After weaning, animals from L12 and L3 were housed three per cage and provided *ad libitum* access to food for 6 months. L3 rats exhibited elevated body weight, along with increased visceral, subcutaneous, and perivascular fat accumulation. However, heart weight at sacrifice was reduced in L3 rats. Furthermore, L3 rats displayed elevated serum levels of glucose, leptin, adiponectin, total lipids, and triglycerides compared to control rats. In the myocardium, overfed rats showed decreased IL-10 mRNA levels and alterations in contractility and heart rate in response to insulin. Similarly, aortic tissue exhibited modified gene expression of TNFα, iNOS, and IL-6. Additionally, L3 aortas exhibited endothelial dysfunction in response to acetylcholine, although insulin-induced relaxation remained unchanged compared to controls. At the molecular level, L3 rats displayed reduced Akt phosphorylation in response to insulin, both in myocardial and aortic tissues, whereas MAPK phosphorylation was elevated solely in the myocardium. Overfeeding during lactation in rats induces endothelial dysfunction and cardiac insulin resistance in adulthood, potentially contributing to the cardiovascular alterations observed in this experimental model.

## 1. Introduction

The prevalence of obesity and its associated health complications has alarmingly increased worldwide, becoming a significant challenge for health systems. The World Health Organization (WHO) stated that the worldwide obesity prevalence approximately tripled between 1975 and 2016, with almost 13% of the world’s adult population being obese in 2016 [1]. In Europe, the WHO regional obesity report stated that in 2022, almost 60% of adults were overweight or obese [2], a trend that is predicted to keep growing in the next years [3].

Several authors and experts have pointed out that one of the most worrying causes of this increase is the high and growing prevalence of childhood obesity, a condition that is associated with the development of obesity-related complications in the long term, such as metabolic and cardiovascular diseases [4,5]. In this regard, it is reported that nutritional alterations in early life, including overnutrition, are associated with metabolic disorders in adulthood [6] due to permanent changes in the central and peripheral systems involved in energy homeostasis [7]. This phenomenon was first proposed by Anders Fordsdahl in the “Adaptative phenotype theory”, which suggested that in response to an unbalanced nutritional environment during the perinatal period, either overfeeding or undernutrition, there are subsequent metabolic accommodations in the fetus/offspring [8,9]. This phenomenon has been widely studied in rodents, in which malnutrition during early life induces crucial changes in hypothalamic structures like the arcuate nucleus, affecting the response to metabolic hormones such as leptin or insulin [10] and producing an impairment in the information about the nutritional status that reaches the central nervous system [11,12].

It is reported that one of the most relevant consequences of overnutrition and obesity during early life is the development of insulin resistance [6], a condition associated with a decreased activation of the PI3K-Akt pathway in insulin-dependent tissues such as the liver skeletal muscle and adipose tissue, which results in an impaired uptake and storage of glucose, promoting hyperglycemia in the long term [13]. Moreover, since insulin also triggers a wide range of cardiovascular effects under physiological conditions, insulin resistance not only affects metabolic tissues but also the cardiovascular system [14].

In blood vessels, insulin exerts a vasodilator effect [15] and contributes to the maintenance of endothelial function [16]. This effect is produced through the binding of insulin to its receptor in both vascular endothelial cells and vascular smooth muscle cells (VSMCs), which activates the PI3K-Akt pathway and promotes the phosphorylation of nitric oxide synthase (NOS) and the release of nitric oxide (NO) [17]. Moreover, in VSMCs, insulin directly decreases contractility by decreasing intracellular calcium concentration through voltage-sensitive calcium channels [18]. However, insulin also activates the mitogen-activated protein kinase (MAPK) pathway, which mediates the expression of adhesion molecules such as VCAM-1 or E-selectin in the vascular endothelium and the release of the vasoconstrictor peptide endothelin-1 (ET-1) [17]. In physiological conditions, the activation of the PI3K-Akt pathway predominates over the activation of the MAPK pathway, so the release of NO predominates over the release of ET-1. However, in the context of insulin resistance, there is an impaired activation of the PI3K-Akt pathway, whereas the activation of the MAPK pathway remains unaffected, thus producing a decrease in insulin-induced vasodilation and contributing to the development of hypertension in the long term [17,19].

In the heart, insulin is also reported to exert vasodilating effects on coronary arteries, producing an increase in myocardial blood flow [20]. Furthermore, in cardiomyocytes, insulin stimulates glucose uptake and glycogen synthesis, exerts anti-apoptotic effects, and increases heart contractility through the activation of the PI3K pathway [17]. The positive inotropic effects of insulin are associated with the entrance of Ca^2+^ into cardiac myocytes through voltage-dependent Ca^2+^ channels and reverse Na^2+^/Ca^2+^ exchange. Moreover, the influx of Ca^2+^ stimulates an additional release of Ca^2+^ from the sarcoplasmic reticulum via ryanodine receptors, which results in myofilament activation and contraction [17]. Thus, under insulin resistance conditions, there is an impairment in the cardiac effects of insulin producing, among other consequences, reduced contractility.

Finally, despite the described effects of insulin in arterial and myocardial functions, insulin also exerts proliferative effects both in VSMCs and in cardiomyocytes through the activation of the MAPK pathway, so in the context of insulin resistance, the maintained activation of this pathway may lead to the thickening of the arterial wall and to ventricular hypertrophy [21].

In a previous study, our group reported that overfeeding during lactation was associated with both cardiac and vascular insulin resistance in rats at weaning [22]. However, the effects of early overnutrition on cardiovascular insulin sensitivity in the long-term had not been explored yet. Thus, the aim of this study was to evaluate the effects of early overnutrition in the development of cardiovascular insulin resistance in adult male Sprague Dawley rats.

## 2. Results

### 2.1. Body and Organ Weight

Body weight was measured at birth, at weaning (PDN21), and at the age of 6 months, just before euthanasia. At birth, no significant differences were found between rats from control (6.70 g ± 0.73) and reduced litters (7.02 g ± 0.50). However, a significant increase in body weight was found in L3 rats compared to those raised in L12 both at PND21 and at the age of 6 months (*p* < 0.05) (Figure 1A), as well as higher food intake in L3 rats (Figure 1B). Likewise, a significant increase in the weights of visceral and subcutaneous fat, PVAT, soleus, and the heart was also observed (*p* < 0.05) (Table 1).

### 2.2. Glycemia, Lipid Profile, and Plasma Concentrations of Metabolic Hormones: Insulin, Leptin, and Adiponectin

Serum glucose levels were significantly upregulated in overfed rats compared to controls, whereas insulin levels were unchanged between experimental groups (Figure 2A,B) (*p* < 0.05). Leptin (Figure 2C) and adiponectin (Figure 2D) circulating levels were significantly increased in the serum of L3 rats. Regarding lipid profile, total cholesterol serum levels remained unchanged (Figure 2E) when L3 and L12 groups were compared; meanwhile, c-HDL (Figure 2F) and c-LDL (Figure 2G) were significantly lower in overfed rats compared to controls. However, the serum levels of total lipids (Figure 2H) and triglycerides (Figure 2I) were significantly increased in rats from L3 in comparison to rats from L12.

Additionally, the HOMA-IR was calculated, showing that L3 rats had a higher score than L12 (0.9 L12 vs. 1.41 L3), which led to an expectation of decreased insulin sensitivity in overfed rats.

### 2.3. Long-Term Cardiovascular Effects of Early Overnutrition

#### 2.3.1. Myocardial Changes Triggered by Litter Reduction: mRNA Expression, Hemodynamic Parameters, and Activation of Molecular Pathways

The myocardial expression of different genes related to both inflammation and oxidative stress are shown in Table 2. There were only statistically significant differences in IL-10 mRNA expression.

Regarding cardiac function, no significant changes were found between L12 and L3 rats at basal conditions. However, insulin administration promoted a decrease in cardiac contractility (dP/dT) at 10^−9^ M and 10^−8^ M concentrations in the heart from L3 rats in comparison to controls (Figure 3A), as well as in heart rate at 10^−7^ M (Figure 3B). Regarding coronary pressure, no changes were observed at any concentration of insulin (Figure 3C).

When evaluating the two primary pathways activated by insulin in the heart through Western blot analysis, we observed the activation of the PI3K/Akt pathway (*p* < 0.05) in response to insulin in both L3 and L12 rats although, activation of the Akt/PI3K pathway was notably diminished in overfed rats after insulin administration (Figure 4A). Conversely, the MAPK pathway was exclusively triggered by insulin in overfed rats (Figure 4B). 

#### 2.3.2. Arterial Changes Induced by Litter Reduction: mRNA Expression, Hemodynamic Parameters, and Activation of Molecular Pathways

Different from what occurred in myocardial tissue, the expression of TNFα and IL-6, both related to a proinflammatory phenotype, was significantly higher in the aortas of L3 rats in comparison to those of L12 rats (Table 3). Additionally, iNOS gene expression was decreased in overfed rats’ aortas (*p* < 0.05).

Regarding vascular function, statistical differences were found in response to acetylcholine (Ach) (Figure 5A); as well, no differences were found in response to nitroprusside (NTP) (Figure 5B), showing that aortas from overfed rats had lower endothelium-dependent relaxation.

Furthermore, in order to assess insulin resistance in the aorta, the vascular reactivity of aorta segments from L12 and L3 rats was also performed in response to increasing concentrations of insulin. No statistical differences were observed when comparing both groups (Figure 6A), but after preincubation with L-NAME, L12 segments showed decreased relaxation in response to insulin, whilst L3 aortic segments did not (Figure 6B). This implies the molecular pathway involved in the vasodilatory aortic response to insulin in control rats is related to nitric oxide; meanwhile, segments from overfed rats may comprise another one that is not NO-dependent. Several experiments using other blockers were also performed in order to assess the molecular pathways responsible for this difference, but no significant results were obtained.

Similarly to the myocardium, the two main pathways triggered by insulin in the aorta were analyzed, showing activation of PI3K/Akt in response to insulin only in aortas from control rats (Figure 7A), while MAPK remained unaltered (Figure 7B). Contrarily, aortas from the L3 group did not show neither activation of the Akt/PI3K pathway nor of MAPK (Figure 7A,B).

## 3. Discussion

The prevalence of insulin resistance is greatly increasing worldwide, and obesity is the main risk factor for this condition, which, in the long term, is associated with the development of cardiovascular diseases [23]. In this study, we show for the first time that early overnutrition in rats during the lactation period is associated with endothelial dysfunction and cardiac insulin resistance in adulthood, a condition that may contribute to the impairment of cardiovascular function associated with overweight and obesity.

Many studies have reported that overnutrition during lactation in rodents not only results in several metabolic disorders but also in cardiovascular alterations that include cardiac dysfunction and the development of hypertension [24,25,26]. The metabolic changes associated with postnatal overfeeding have been widely described [27] and are due to alterations both in the central systems involved in food intake regulation [28,29,30,31] and in peripheral tissues involved in metabolism such as the gastrointestinal system, the liver, and the adipose tissue [32,33]. In this regard, our results agree with previous studies, which have reported that early overnutrition is associated with increased body weight at weaning, which is maintained in adulthood [22,34,35]. This increased body weight is the result of increased adiposity, both at the visceral and at the subcutaneous level, probably as a result of adipocyte hypertrophy and hyperplasia, which is translated into a higher capacity of adipose tissue for energy storage [36]. Moreover, increased adiposity in L3 rats was also associated with increased circulating levels of leptin as previously described [27]. However, despite hyperleptinemia, it is known that early overnutrition in rats is associated with leptin resistance [37], which is probably the reason why overfed rats are hyperphagic compared to control animals.

Additionally, our findings demonstrate that excessive feeding during the lactation period in rats leads to increased glycemia in adulthood but without producing neither hyperglycemia nor hyperinsulinemia. Nevertheless, when assessing insulin resistance through the homeostasis model assessment for insulin resistance (HOMA-IR index), we observed reduced insulin sensitivity in overfed rats compared to controls. These results agree with those reported by other researchers, indicating that glucose levels tend to rise both in the short and long term in rats subjected to overnutrition during the lactation period [38,39]. It is plausible that hyperinsulinemia in the short term [22] occurs as a compensatory mechanism in response to early overnutrition. However, in the long term, this effect seems to be attenuated, although reduced insulin sensitivity persists in adulthood. As a result, the higher glucose levels in overfed rats may be attributed, at least in part, to increased food intake and/or decreased insulin sensitivity. On the other hand, early overfed rats showed increased circulating levels of adiponectin compared to controls. This finding is consistent with a previous study, which reported that the litter reduction model is associated with increased circulating levels of adiponectin due to increased adiposity [40]. This observation holds significance as adiponectin plays a crucial role in enhancing insulin sensitivity by promoting fatty acid oxidation and inhibiting hepatic glucose production [38]. Thus, although early overnourished rats showed increased HOMA-IR, the increased adiponectin levels may be related to the absence of both hyperglycemia and hyperinsulinemia.

In addition to increased adiposity and reduced insulin sensitivity, our results show that postnatal overfeeding is also associated with significant alterations in the lipid profile. Particularly, rats raised in reduced litters showed reduced circulating levels of HDL-c and LDL-c but increased serum concentrations of total lipids and triglycerides, which agrees with previous studies in experimental models of early overnutrition [41,42]. Moreover, these results are in line with other experimental rat models of obesity based on a high-fat diet [43,44]. In these models, diet-induced obesity is associated with increased circulating levels of cholesterol, LDL-c, and triglycerides, and these alterations are improved with the withdrawal of the high-fat diet or by subjecting the animals to a caloric restriction. However, in our model, changes in glycemia and lipid profile start at an early age [22] and remain in the long term despite feeding the animals with a standard diet. This fact demonstrates that the changes in central and peripheral pathways induced by early overnutrition have crucial importance on the potential development of metabolic alterations in adulthood.

As previously described [26], our results also show that, in addition to the metabolic disturbances, overfeeding during lactation in rats was associated with cardiovascular alterations both at the vascular and the cardiac levels. In the myocardium, overnourished rats showed decreased heart weight compared to controls, with this effect being possibly due to an increase in cardiomyocytes’ apoptosis as it has been previously reported [35]. However, this result is controversial since other studies have reported either no changes [27,45] or even an increase in heart weight together with increased cardiomyocyte area [46,47,48,49]. These results may be due, at least in part, to the different species used (rats and mice), the different number of animals in control (8 to 12) and reduced litters (3 to 4), and the different ages of the animals.

In terms of cardiac function, our results show decreased heart contractility in response to insulin administration in overfed rats, which denotes the existence of cardiac insulin resistance. This decrease in myocardial insulin sensitivity in response to overfeeding during lactation has already been reported in 21-day-old rats [22] and in adult mice [48,50]. In both cases, early overnutrition produced a decrease in the activation of the PI3K-Akt pathway in myocardial tissue, which, in the rat model, was also translated in decreased heart contractility in response to insulin [35]. Moreover, in overfed rats, the decreased myocardial contractility was associated with a higher heart rate after insulin administration, which may constitute a compensatory mechanism to balance the lower contractility. This study then corroborates that the cardiac insulin resistance induced during early life in response to overfeeding during lactation is maintained in adulthood in rodents, with this fact affecting cardiac function. Furthermore, as previously described in the short term [22], our results show that cardiac insulin resistance is not only associated with impaired activation of the PI3K-Akt pathway but also with overactivation of the MAPK pathway in response to insulin. The PI3K-Akt axis is reported to play a key role in maintaining adequate cardiac function and offering protection against various pathologies, including myocardial infarction [51]. In contrast, the activation of the MAPK pathway induces inflammation and results in the development of cardiac damage [52]. Thus, in addition to cardiac insulin resistance, the imbalance between the PI3K/Akt and the MAPK pathways may have implications for myocardial function, making the heart more susceptible to potential pathologies. Indeed, in our model, there is a decrease in the gene expression of IL-10 in hearts from L3 rats, which may be a signal of the proinflammatory status due to the MAPK overactivation and/or to the increased heart rate, which also produces a micro-inflammatory state [53].

In the aorta, early overfed rats also showed increased gene expression of the pro-inflammatory cytokines IL-6 and TNFα compared to control rats, possibly as a result of increased infiltration of macrophages into the arterial tissue as is reported in other obesity models [54,55]. The increased arterial inflammation has been also described in 21-day-old rats [22] and is associated with impaired endothelium-dependent relaxation in response to acetylcholine without changes in endothelium-independent relaxation in response to Sodium Nitroprusside (SNP). The endothelial dysfunction associated with postnatal overfeeding in rodents has been reported before and is present both in the long [47] and the short term [22]. However, other authors did not find changes in acetylcholine-induced relaxation [27], indicating that changes in vascular function in rats raised in reduced litters might be influenced by different factors such as the strain, the sex, or the age of animal models.

Regarding vascular insulin resistance, segments from L3 adult rats did not exhibit changes in insulin-induced relaxation compared to segments from L12 adult rats, pointing to intact vascular insulin sensitivity. This result suggests that although early overnutrition induces vascular insulin resistance in the short term [22], in the long term, the vascular system may compensate for this situation by activating other molecular pathways during growth, thereby preserving insulin sensitivity. At the molecular level, opposite than in the heart, our results show decreased activation of the PI3K/Akt pathway in the arterial tissue of L3 rats in response to insulin but without changes in the activation of the MAPK pathway. Thus, it is possible that the development of insulin resistance requires not only impaired PI3k/Akt signaling but also hyperactivation of the MAPK pathway.

## 4. Materials and Methods

### 4.1. Animals

This study was performed using 6-month-old Sprague Dawley male rats. All of the experiments were conducted according to the European Union Legislation (RD53/2013) and with the approval of the Animal Care and Use Committee of the Community of Madrid (Spain).

The experimental model was based on litter reduction, a widely used childhood obesity model in rodents [32]. Briefly, there were two groups based on the number of litters left after birth: 12 pups (L12) or 3 pups (L3) per mother. After this initial adjustment, the offspring did not undergo any further manipulation during the lactation period. After weaning, male rats were housed 3 per cage and fed ad libitum and under normal conditions (50–60% humidity, 22–24 °C, and cycles of 12 h light/darkness). Body weight gain and food intake were assessed weekly. At the age of 6 months, rats were euthanized after intraperitoneal injection of sodium phentobarbital (100 mg/kg) after overnight fasting. The blood was collected to obtain the serum, and tissues were dissected and weighed.

### 4.2. Plasma Measurements

Before sacrifice, glycemia was measured using a glucometer in both L3 and L12 rats. The serum measurements of total cholesterol, c-HDL, c-LDL, triglycerides, and total lipids were made using commercial kits of SPINREACT^®^ (Spinreact, Girona, Spain) following the manufacturer’s instructions. The analysis of insulin, leptin, and adiponectin levels was performed through ELISA (Enzyme-Linked ImmunoSorbent Assay) using a commercial kit from Millipore (Dramstadt, Germany).

### 4.3. Experiments of Vascular Reactivity

Immediately after sacrifice, aortas from both L3 and L12 rats were dissected, cut into 2 mm segments, and placed into a tension myograph system. Briefly, the process consisted of a chamber filled with 4 mL of 370C Krebs-Henseleit buffer (115 mM NaCl, 4.6 mM KCl, 1.2 mM KH_2_PO_4_, 1.2 mM MgSO_4_, 2.5 mM CaCl_2_, 25 mM NaHCO_3,_ and 11 mM glucose) and oxygenated (95% O_2_ and 5% CO_2_). The changes in isometric force were recorded using an isometric tension recording (Universal Transducing Cell UC3 and Statham Microscale Accessory UL5, Statham Instruments, Inc., Los Angeles, CA, USA) and PowerLab data acquisition system (ADInstruments, Colorado Springs, CO, USA), just like previously described [50].

Dose–response curves to increasing concentrations of acetylcholine (Ach) (10^−9^–10^−5^ M) and sodium nitroprusside (NTP) (10^−9^–10^−5^ M) were determined to assess relaxing and contractile responses. Moreover, the response to insulin was also evaluated using increasing and accumulative doses of insulin (10^−8^–10^−5^ M). In some cases, several segments were preincubated for 20 min with Nω-Nitro-L-arginine methyl ester (L-NAME) (10^−4^ M) (Sigma-Aldrich, St. Louis, MO, USA), an unspecific inhibitor of endothelial nitric oxide synthase (eNOS), Apamin (10^−6^ M), and charibdotoxin (10^−7^ M) (Sigma-Aldrich, St. Louis, MO, USA), which are blockers of Calcium-dependent potassium channels, or with ouabain (10^−6^ M) (Sigma-Aldrich, St. Louis, MO, USA), a blocker of the sodium–potassium pump, in order to assess the pathways involved in relaxation processes.

### 4.4. Experiments of Cardiac Function: Langendorff

After sacrifice, hearts from both L3 and L12 were extracted and placed in a Langendorff system. Briefly, heparin (1000 IU) was injected intravenously, and the heart was immediately cannulated through the ascending aorta, producing a retrograde perfusion of the coronary circulation. The heart was kept in a bath at 37 °C, perfused with Krebs-Henseleit buffer, and equilibrated with 95%/5% oxygen and carbon dioxide. Then, perfusion was initiated at a flow rate of 6–8 mL/min, producing a basal perfusion pressure of around 70 mmHg). In order to record the intraventricular pressure, a small plastic balloon connected to a Statham transducer (Statham Instruments, Los Angeles, CA, USA) was inserted into the left ventricle. The balloon was inflated to a diastolic pressure of 5–10 mmHg. From the intraventricular pressure, we were able to calculate the first derivative with respect to time (dP/dT), which is the main indicator of heart contractility and heart rate. All of these parameters were recorded both before and after the insulin curve (10^−9^–10^−7^ M). At the end of the experiment, hearts were removed from the perfusion system and kept frozen at −80 °C for subsequent molecular analysis.

### 4.5. Nitrite and Nitrate Determination in the Culture Medium

The amount of NO_2_^−^ and NO_3_^−^ was measured in the medium as an indirect indicator of NO release by the endothelium. NO_2_^−^ and NO_3_^−^ concentrations were determined after incubation of 5 mm aorta segments in 24-well plates using DMEN/F-12 + 1% of an antibiotic medium. Some segments were also incubated with insulin 10^−7^ M for 30 min. Then, the culture medium was collected, and NO_2_^−^ and NO_3_^−^ were quantified by using the Griess method [51]. Likewise, aorta segments were also collected to further analyze the activation of the PI3K/Akt and MAPK pathways by using Western blot.

### 4.6. Protein Quantification by Western Blot

Aorta segments incubated in the presence or absence of insulin (10^−7^ M) and hearts used in the Langendorff technique were used to analyze the expression and activation of the intracellular pathways Akt and MAPK in response to insulin. For that purpose, total proteins were extracted from 100 mg of arterial or myocardial tissues by homogenization using RIPA buffer (50 mM HEPES, 150 mM NaCl, 1 mM MgCl_2_, 1 mM CaCl_2_, 10 mM de sodium pyrophosphate, 10 mM NaF, 2 mM EDTA, 10% glycerol, 1% NP-40, and pH 7.4) and a cocktail of protease inhibitors (Roche Diagnostics, Barcelona, Spain). The total protein amount was quantified using the Bradford assay (Merck Millipore, Dramstadt, Germany). Subsequently, each sample containing 100 μg of total protein (95% Laemmli Sample Buffer (Bio-Rad, Hércules, CA, USA) and 5% of 2-mercaptoetanol) (Sigma-Aldrich, St. Louis, MO, USA) was loaded into SDS-PAGE gels (10%). Afterward, proteins were transferred to a PVDF membrane (Bio-Rad, Hércules, CA, USA), blocked with a blocking solution (5% BSA for phosphorylated proteins and 5% non-fat dried milk for non-phosphorylated proteins), and incubated overnight with the appropriate primary antibody at 4 °C. Then, the membranes were incubated with the secondary antibody (1:2000) bound to horseradish peroxidase (HRP) for 90 min at room temperature. Subsequently, membranes were revealed with ECL (BioRad, Hercules, CA, USA) in a BioRad Molecular Imager transistor ChemiDoc XRS System (BioRad, Hercules, CA, USA). The density of the bands was quantified using ImageJ software V1.53.

### 4.7. RNA Extraction and Quantitative RT Real-Time PCR

Total RNA was extracted from 100 mg of arterial and cardiac tissues using the Chomczynski method [52] and quantified with a Nanodrop2000 spectrometer (Thermo Fisher Scientific, Hampton, NH, USA). Afterward, 1µg of total RNA was retrotranscripted into cDNA using a high-capacity cDNA reverse transcription kit (Applied Biosystems, Foster City, CA, USA).

Fluorogenic (FAM, VIC) primers designed by the Assay-onDemandTM mouse gene expression products were used (Applied Biosystems; COX2: Rn00585003_m1; glutathione reductase (GRS): Rn01482159_m1; glutathione peroxidase (GPX3): Rn00574703_m1; iNOS: Rn06325074_m1; tumor necrosis factor Alpha (TNFα): Rn00562055_m1; superoxide dismutase (SOD1): Rn00566938_m1; Interleukin-6 (IL-6): Rn01410330_m1; Interleukin-10 (IL-10): Rn01483987_m1; NADPH oxidase 1 (NOX1): Rn00586652_m1; NOX4: Rn00585380_m1; and 18S: Rn03928990_g1).

### 4.8. Statistical Analysis

All data were represented as the mean (±SEM). Quantitative data between L12 and L3 were compared by using Student’s *t*-test or ANOVA. Every analysis was performed using SPSS v18 software, considered statistically significant when *p* < 0.05.

## 5. Conclusions

Postnatal overfeeding in rats induces cardiac insulin resistance and endothelial dysfunction but is not associated with vascular insulin resistance in the long term. Although a more comprehensive exploration of the molecular pathways involved is warranted, our findings suggest that nutritional imbalances during the early stages of life can initiate changes that predispose individuals to metabolic and cardiovascular disorders in adulthood, so nutrition during early life is an important factor to consider to avoid its possible deleterious effects on health in the long term.

## Figures and Tables

**Figure 1 ijms-24-14443-f001:**
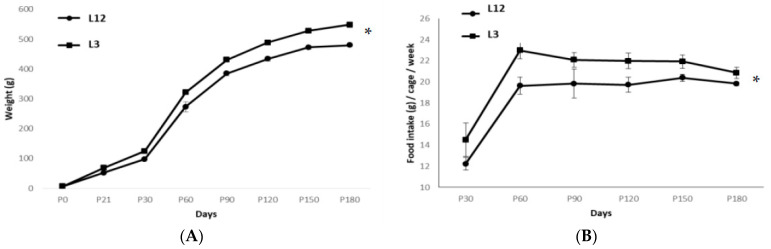
(**A**) Body weight gain over 6 months until sacrifice of control (n = 12) and overfed rats (n = 9) and (**B**) respective mean amount of food intake per cage. Values are represented as Mean ± SD. * *p* < 0.05 vs. L12.

**Figure 2 ijms-24-14443-f002:**
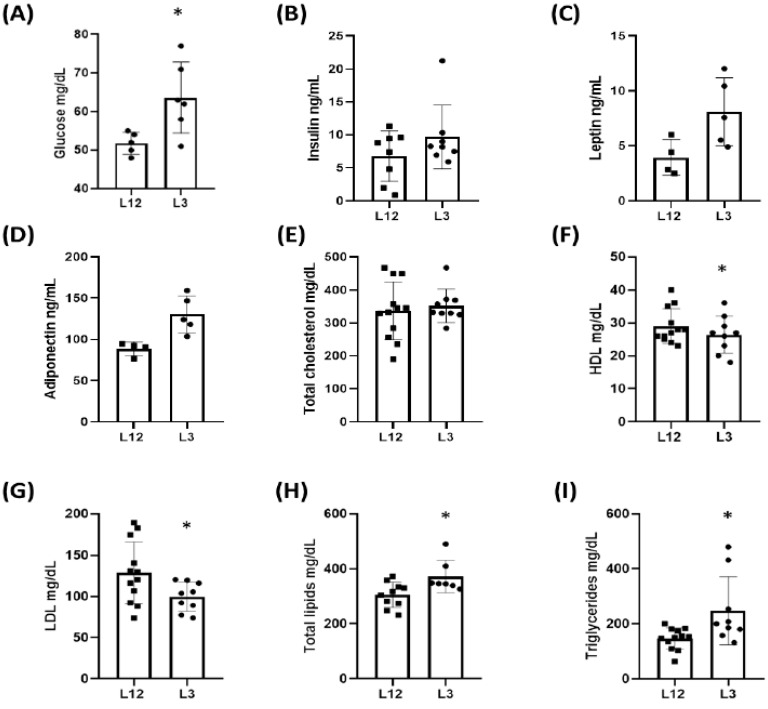
Plasma measurements of glucose (mg/dL) (**A**), insulin (ng/mL) (**B**), Leptin (ng/mL) (**C**), Adiponectin (ng/mL) (**D**), Total cholesterol (mg/dL) (**E**), HDL cholesterol (mg/dL) (**F**), LDL cholesterol (mg/dL) (**G**), Total lipids (mg/dL) (**H**), and triglycerides (mg/dL) (**I**),in control rats (L12) and in overfed rats (L3). Values are expressed as Mean ± SEM. * *p* < 0.05 vs. L12.

**Figure 3 ijms-24-14443-f003:**
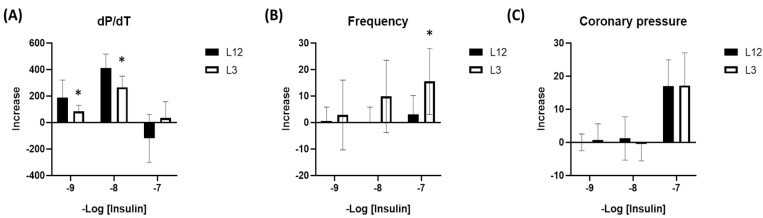
Increment of different parameters measured using Langendorff system related to cardiac function in response to increasing concentrations of insulin (10^−9^–10^−7^ M), including contractility measured as dP/dT (**A**), heart rate (**B**), and coronary pressure (**C**). * *p* < 0.05 vs. L12.

**Figure 4 ijms-24-14443-f004:**
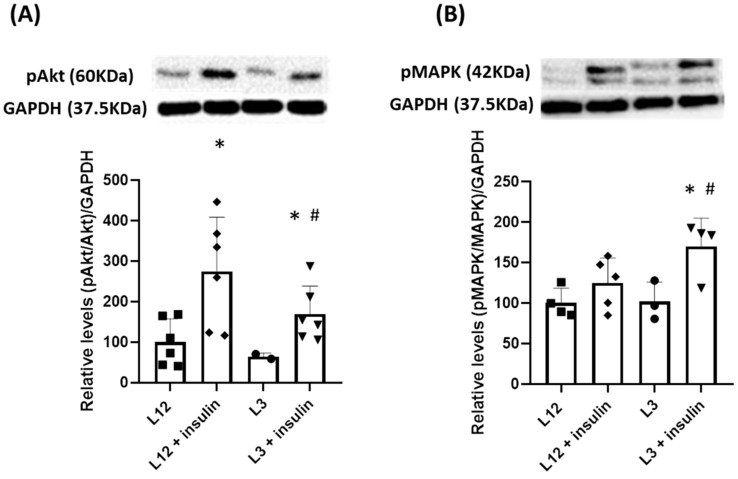
Relative quantification of activation, measured as phosphorylated protein vs. total protein, in myocardium tissue in response to insulin in Akt pathway (**A**) and MAPK pathway (**B**). *, *p* < 0.05 vs. basal; #, *p* < 0.05 vs. L12 + insulin.

**Figure 5 ijms-24-14443-f005:**
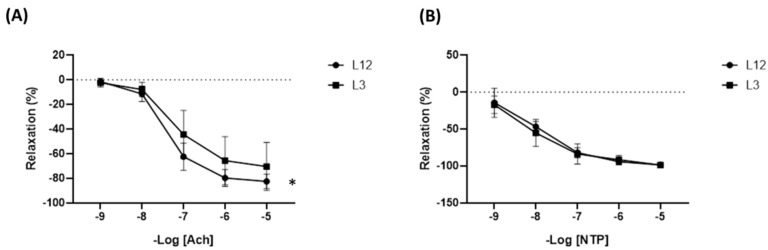
Relaxation curves in response to increasing concentrations of acetylcholine (Ach) (**A**) and nitroprusside (NTP) (**B**) in aortic rings from both L12 and L3 rats. * *p* < 0.05 vs. L12.

**Figure 6 ijms-24-14443-f006:**
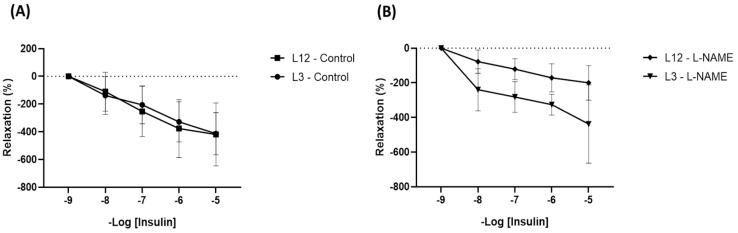
Relaxation curves in response to increasing concentrations of insulin (**A**) and insulin + L-NAME, blocker of nitric oxide (**B**) of aortic rings, both from L12 and L3 rats.

**Figure 7 ijms-24-14443-f007:**
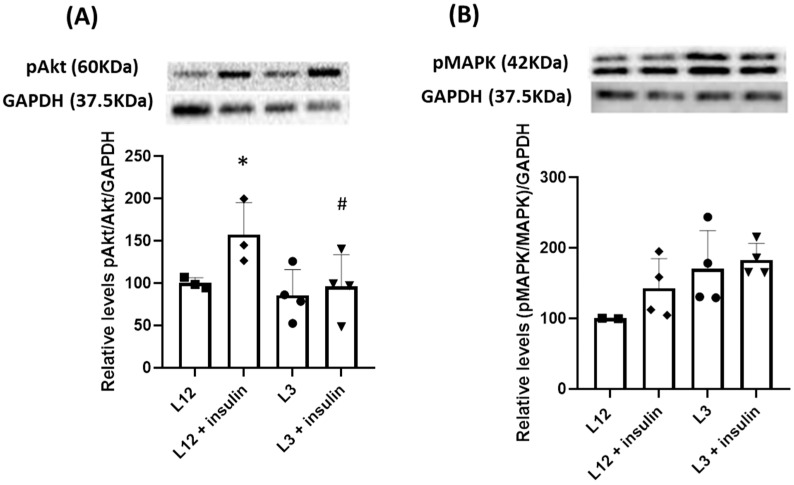
Relative quantification of activation, measured as phosphorylated protein vs. total protein, in aortic tissue in response to insulin in Akt pathway (**A**) and MAPK pathway (**B**). *, *p* < 0.05 vs. basal; #, *p* < 0.05 vs. L12 + insulin.

**Table 1 ijms-24-14443-t001:** Organ weights. Data are presented as mean ± SEM. Control rats (L12): n = 10 and overfeed rats (L3): n = 8. * *p* < 0.05 vs. L12, ** *p* < 0.01 vs. L12, *** *p* < 0.001 vs. L12.

	L12	L3
Visceral Fat	1354.12 ± 51.9	1655.10 ± 110.9 ***
Subcutaneous Fat	504.27 ± 115.1	666.18 ± 143.3 **
Brown Fat	112.50 ± 28.6	115.55 ± 51.1
PVAT	16.26 ± 5.1	21.53 ± 2.2 *
Gastrocnemius	505.90 ± 12.7	491.43 ± 11.8
Soleus	39.50 ± 1.8	45.56 ± 1.4 *
Liver	2435.83 ± 59.3	2354.85 ± 51.2
Heart	432.24 ± 19.7	387.08 ± 10.6 *

**Table 2 ijms-24-14443-t002:** mRNA levels of *COX2*, *GRS*, *GPX3*, *TNFα*, *SOD1*, *iNOS*, *IL-1β*, *IL-6*, *IL-10*, *NOX1*, and *NOX4* in the myocardium of L12 and L3. Every gene was quantified relatively using 18S gene expression levels. Values are expressed as percentage ± SEM. * *p* < 0.05 vs. L12.

Gene	L12 (Basal)	L3 (Gene Expression/18S)
*COX2*	100 ± 20.7	80.30 ± 15
*GRS*	100 ± 18.9	121.95 ± 7
*GPX*	100 ± 7.7	115.45 ± 1.4
*TNFα*	100 ± 17.5	80.68 ± 24.1
*SOD1*	100 ± 9.6	105.13 ± 0.8
*iNOS*	100 ± 44.3	45.21 ± 13.8
*IL-1β*	100 ± 20.1	126.32 ± 32.3
*IL-6*	100 ± 13.9	79.14 ± 3.5
*IL-10*	100 ± 19.4	42.86 ± 18.7 *
*NOX1*	100 ± 34.2	156.31 ± 22.1
*NOX4*	100 ± 30.9	94.14 ± 6.5

**Table 3 ijms-24-14443-t003:** mRNA levels of *COX2*, *GRS*, *GPX3*, *TNFα*, *SOD1*, *iNOS*, *IL-1β*, *IL-6*, *IL-10*, *NOX1*, and *NOX4* in the arterial tissue from L12 and L3 rats. Every gene was quantified relatively using 18S gene expression levels. Values are expressed as percentage ± SEM. * *p* < 0.05 vs. L12.

Gene	L12 (Basal)	L3 (Gene Expression/18S)
*COX2*	100 ± 10.7	114.82 ± 24.6
*GRS*	100 ± 18.1	149.72 ± 43.2
*GPX*	100 ± 12.4	174.25 ± 70.2
*TNFα*	100 ± 27.6	175.17 ± 27.3 *
*SOD1*	100 ± 30.6	131.69 ± 34.1
*iNOS*	100 ± 29.5	24.4 ± 10.6 *
*IL-1β*	100 ± 25.1	185.93 ± 56.9
*IL-6*	100 ± 17.5	191.78 ± 14.7 *
*IL-10*	100 ± 33.9	84.52 ± 15.8
*NOX1*	100 ± 56	118.88 ± 66.5
*NOX4*	100 ± 21.3	89.84 ± 11.3

## Data Availability

Data not available due to privacy.

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
