# Peer review of "Postnatal Overfeeding during Lactation Induces Endothelial Dysfunction and Cardiac Insulin Resistance in Adult Rats"

_ijms, 2023, doi:10.3390/ijms241914443_

Round 1

Reviewer 1 Report

Animal models provide the unique opportunity to investigate mechanical aspects of cardiovascular system function, but essential caveats exist when extrapolating findings obtained from preclinical models of diabetes to humans. Indeed, animal models do not recapitulate the complexity of environmental factors, most notably the duration of the exposure to insulin resistance that may play a crucial role in the development of cardiovascular system dysfunction. Moreover, most preclinical studies are performed in animals with uncontrolled or poorly controlled diabetes, whereas patients tend to undergo therapeutic intervention. Finally, while type 2 diabetes mellitus prevalence trajectory mainly increases at 40- < 75 years (with a currently alarming increase at younger ages), it is a legitimate concern how closely rodent models employing young animals recapitulate the disease developing in old people. 

In the current study, the written mean value of glucose is 70 mg/dl in the L3 group (Figure 2), which is not hyperglycemic, as reported in line 249. Moreover, all obese animals without DM have increased insulin levels compared to lean animals. So, obese animals need more insulin than lean. So, an adjustment for insulin levels should be performed before the conclusion of insulin resistance. 

Therefore, I need to find out if the title of the manuscript is correct.

The methodology is appropriate. The manuscript is well written, but the discussion/conclusions should be better. 

The data are interesting, but the study title should be changed.

none

Author Response

As suggested by the reviewer, the discussion and conclusions have been extensively improved. All changes are in red color.

Following the suggestion of the reviewer we have changed the title of the manuscript to:

“Postnatal overfeeding during lactation induces endothelial dysfunction and cardiac insulin resistance in adult rats”

Reviewer 2 Report

Dear Editor,

thanks so much for the opportunity to revise the work entitled "Postnatal overfeeding during the lactation period in rats induces cardiovascular insulin resistance in adulthood”. The work is very interesting, showing that early overnutrition is associated with cardiometabolic alterations in adulthood, likely attributed to reduced insulin sensitivity due to its crucial role in the cardiovascular  system. The paper is well written, the results are clearly reported and the literature is rigorous.

Introduction could more clearly demonstrate the gap in the literature and tell the story. In particular, I feel like the justification for investigate relationship between postnatal overfeeding and insulin resistance was poorly described.

The results section is detailed, there is no need to change anything.

I also suggest improving the discussion. Prevalence of insulin resistance worldwide could be described in further depth. 

Thanks.

Author Response

Introduction could more clearly demonstrate the gap in the literature and tell the story. In particular, I feel like the justification for investigate relationship between postnatal overfeeding and insulin resistance was poorly described.

As suggested by the Reviewer the introduction has been rewritten to highlight the importance of insulin in cardiovascular system and the relationship between postnatal overfeeding and insulin resistance. All changes are in red color.

The results section is detailed, there is no need to change anything.

I also suggest improving the discussion. Prevalence of insulin resistance worldwide could be described in further depth.  Thanks.

As suggested by the reviewer, we have rewritten the discussion section and, in our opinion, it has been extensively improved. All changes are in red color.

Round 2

Reviewer 1 Report

no other comments

none